# External Electric Field Tailored Spatial Coherence of Random Lasing

**Yaoxing Bian [1,2], Hongyu Yuan [1], Junying Zhao [1], Dahe Liu [1], Wenping Gong [1,*] and Zhaona Wang [1,*]**

1    Applied Optics Beijing Area Major Laboratory, Department of Physics, Beijing Normal University, Beijing 100875, China
2    College of Physics and Optoelectronics, Taiyuan University of Technology, Taiyuan 030024, China
*    Correspondence: wpgong@bnu.edu.cn (W.G.); zhnwang@bnu.edu.cn (Z.W.)

**Abstract:** In this study, spatial coherence tunable random lasing is proposed by designing a random laser with separate coupling configuration between the gain medium and the scattering part. By using the polymer dispersion liquid crystal (PDLC) film with tunable scattering coefficient for supplying random scattering feedback and output modification, red, green and blue random lasers are obtained. By applying or removing electric field to manipulate the scattering intensity of the PDLC film, intensity and spatial coherence of these random lasing are then switched between the high or low state. This work demonstrates that controlling the external scattering intensity is an effective method to manipulate the spatial coherence of random lasing.

**Keywords:** random laser; spatial coherence; random scattering feedback; tunable scattering coefficient

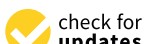



## 1. Introduction

As an unconventional laser, random lasing with low spatial coherence and high brightness can effectively suppress speckle noise [1], and plays a vital role in the fields of optical imaging [2–4], display [5], sensing [6,7], information security [8,9] and integrated photoelectron [10]. The optical field of random lasing is endowed with rich physical characteristics due to the inherent randomness of the multiple-scattering feedback mechanism [11–13]. Therefore, the optical field manipulation of random lasing has become an important research topic [14–16]. In particular, low spatial coherence is the unique characteristics of random lasing different from the high spatial coherence of traditional lasing. Regulating the spatial coherence of random lasing is conducive to broaden its application fields [17,18]. At present, researchers have regulated the spatial coherence of random lasing by changing the scattering mean free path [19], pump area [20] and pump power density [21]. The method of controlling the scattering mean free path is widely used for tuning random lasing by changing the scatterer concentration in the gain medium due to its simple operation [19]. However, this manipulation method is not conducive to dynamically regulating the spatial coherence of random lasing because of the configuration of directly mixing the gain medium and the scattering material together. Recently, the separate coupling of individual gain medium and individual scattering part has been reported as a new configuration for flexibly regulating the random lasing pulses in the time domain [22], demonstrating a huge superiority in dynamic regulation. As a new coupling configuration, the effect of external scattering intensity on the random lasing has not been reported. The potential of this separate coupling configuration in dynamically regulating the spatial coherence of random lasing needs to be further revealed. Therefore, it is necessary to propose a random laser with a separate coupling configuration to realize spatial coherence tunable random lasing through dynamically adjusting the scattering coefficient of the scattering part.

In this work, a random laser with dynamically tunable spatial coherence is proposed based on the separate coupling between gain medium and scattering part. The PDLC film with adjustable scattering coefficient is coupled with different dye solutions for colorful

random lasing. The coherent feedback random lasing with wavelengths of 457.5 nm, 542.7 nm and 662.9 nm are realized. Furthermore, the effect of the scattering coefficient on the optical field of random lasing has also been studied. A small scattering coefficient means a high intensity random lasing with high spatial coherence. Therefore, the intensity and spatial coherence of random lasing can be flexibly manipulated by applying or removing the electric field on the PDLC film. This work can further expand the application of random lasing in dynamic imaging, display and other fields.

## 2. Materials and Methods

The gain dyes selected in the experiments were Coumarin 1(C1, ACROS), Coumarin 153 (C153, Tokyi Chemical Industry, Tokyo, Japan) and 4-(dicyanomethylene)-2-tert-butyl-6-(1,1,7,7-tetramethyljulolidin-4-yl-vinyl)-4H-pyran (DCJTB, Tokyi Chemical Industry, Tokyo, Japan). Firstly, these gain materials were diluted in ethanol to obtain the C1 solution with a concentrate of 1.0 mg mL$^{-1}$, the C153 solution at 1.5 mg mL$^{-1}$ and the DCJTB solution at 1.5 mg mL$^{-1}$ for further operation. Then, 1.50 mL dye solution was added into the quartz cuvette as the gain part. Finally, the commercial PDLC film with a thickness of 400 μm was coupled to the outside of the quartz cuvette as the tunable scattering part of the random laser, and the average size of LC droplets was about 400 nm.

The schematic diagram and optical photo of the experimental setup are shown in Figure 1a,b, respectively. The samples were optically excited by a frequency-doubled and Q-switched neodymium doped yttrium aluminum garnet (Nd:YAG) laser with a wavelength of 355 nm, repetition rate of 10 Hz, and pulse duration of 8 ns. The pump power density was controlled by an adjustable attenuation element. The dye solution was pumped vertically from the top of the quartz cuvette by the pulses with a spot diameter of 8.0 mm. The scattering coefficient of the PDLC film was regulated by applying and removing the AC voltage of 40 V on the PDLC film. The spectra of emission light passing through PDLC film were measured by a spectrometer of Ocean Optics model Maya Pro 2000 with a spectral resolution of 0.4 nm and an integration time of 100 ms. All experiments were performed at room temperature.

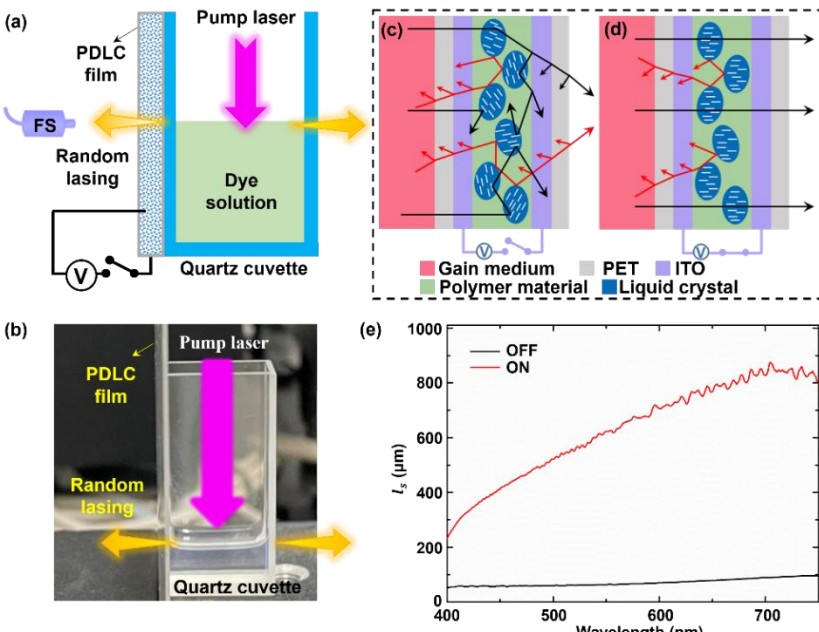

**Figure 1.** Schematic diagram of spatial coherence-tunable random lasing. (**a,b**) The schematic diagram (**a**) and the optical photo (**b**) of the experimental setup. (**c,d**) The emission principle diagram of random lasing in the coupling system with the gain medium and the PDLC film without (**c**) and with (**d**) the external electric field. (**e**) The calculated scattering mean free path ($l_s$) when the electric field was switched on (ON) and switched off (OFF), respectively.

## 3. Results and Discussion

PDLC is widely applied in optical devices due to the excellent tuning possibilities and extreme form of anisotropic scattering [23]. PDLC film is a composite film made by mixing nematic liquid crystal (LC) and polymer in a certain proportion, which is sandwiched between two transparent conductive films with Polyethylene terephthalate (PET) and Indium tin oxide (ITO). The refractive index difference between PET and ITO films and between ITO and air enables them to reflect the incident light. Here, the used LC was an anisotropic material with refractive index $n_o$ (ordinary) and $n_e$ (extraordinary) for further tuning the scattering coefficient of the PDLC film through external bias voltage. The mixed polymer with refractive index $n_p$ ($n_o \approx n_p$) was used to form the PDLC film. Approximately, for a LC droplet with an orientation angle $\theta$ of LC molecule relative to the light polarization direction, the equivalent refractive index $n_{eff}$ can be expressed as $\frac{1}{n_{eff}^2} = \frac{\cos^2\theta}{n_o^2} + \frac{\sin^2\theta}{n_e^2}$ [24,25]. The mismatch of refraction index between the LC droplet and polymer can be written as $\Delta n = n_{eff} - n_o$, which can be controlled by the LC molecule orientation. Therefore, the scattering coefficient of PDLC film can be controlled through using the electric field to change the LC orientation and then induce the mismatch of refractive index between polymer and LC.

The design principle of the spatial coherence tunable random lasing is schematically presented in Figure 1c,d by coupling the mentioned PDLC film and gain region with varying dye solution. When the gain medium is pumped, the emission photons are radiated from the gain region. Some of them are reflected and scattered to the gain region by the PDLC film, supplying random feedback for the generation of random lasing. Some of the emission photons pass through the PDLC film, superimposing additional modification in random phase and intensity. The emitted random lasing is thus obtained and its performance can be modified by the scattering property of the PDLC film. When the PDLC film is without electric field (Figure 1c, OFF), the orientation of LC molecules is highly random, inducing strong scattering due to the highly mismatched refractive index between polymer and LC droplet. Thus, the PDLC film presents an opaque state, demonstrating a small scattering mean free path ($l_s$) obtained through measuring the extinction coefficient [26] as shown in Figure 1e. This PDLC film in OFF state supplies a strong backscattering feedback for the gain part and strong tuning capability of emission light in spatial coherence. Thus, the emitted random lasing has low intensity and spatial coherence. When the external electric field is applied to the PDLC film (Figure 1d, ON), the orientation of LC molecules is almost consistent with the direction of electric field. The corresponding PDLC film is transparent and has a relatively large scattering mean free path (Figure 1e). The weak backscattering feedback is for the gain part and a slight decrease acts on the spatial coherence of the forward emission light. As a result, the emitted random lasing has high intensity and spatial coherence. Therefore, the spatial coherence of the proposed random lasing can be flexibly regulated by switching ON/OFF the applied electric field.

The effect of scattering feedback of individual scattering part on random lasing is demonstrated by measuring the emission spectra of the random lasers under different pump power densities in Figure 2. A blue random laser was obtained by coupling the C1 solution and PDLC film, and named as $RL_B$. The emission spectra of $RL_B$ at different pump power densities when the external electric field is removed from the PDLC film in OFF state are shown in Figure 2a. Only broadband fluorescence spectra can be observed as the pump power density is less than 0.30 MW cm$^{-2}$. However, when the pump power density is greater than 0.30 MW cm$^{-2}$, many narrow linewidth peaks can be observed around 457.5 nm. The results show that the emission light from the C1 solution is randomly scattered by the PDLC film and the coherent resonance is established in the coupling system. Meanwhile, the variations of integral intensity of the emission spectra with pump power density are shown in Figure 2c. When the pump power density is greater than 0.30 MW cm$^{-2}$, the integral intensity increases sharply with increasing pump power density, which indicates that the threshold of the random lasing is about 0.30 MW cm$^{-2}$ [27,28].

The emission spectra of $RL_B$ at different pump power densities when the electric field is applied to the PDLC film in ON state are shown in Figure 2b. Coherent feedback random lasing can also be obtained when the pump power density is greater than 0.20 MW cm$^{-2}$, and the intensity is much higher than that of PDLC film in OFF state. The corresponding threshold of $RL_B$ is about 0.20 MW cm$^{-2}$ (Figure 2c), which is smaller than that of the PDLC film in OFF state. The increase in intensity and decrease in threshold are due to the weak scattering feedback of the PDLC film in ON state.

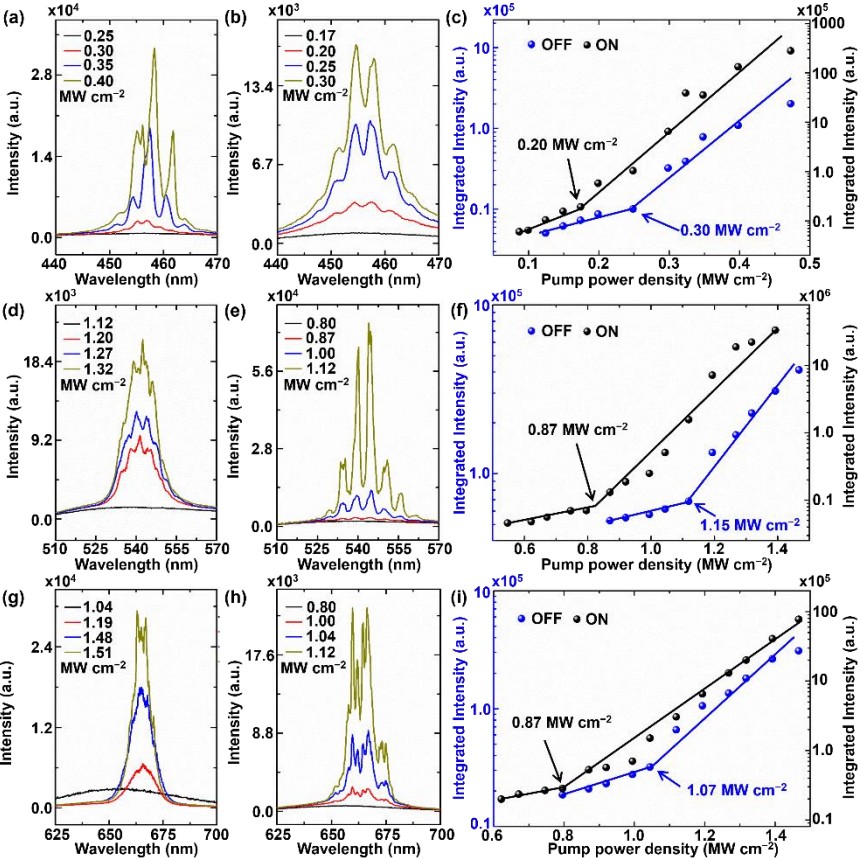

**Figure 2.** Performance characterization of the random lasers. (**a**,**b**) The emission spectra of $RL_B$ at different pump power densities when the electric field is removed (**a**) and applied (**b**). (**c**) The integral intensity varies with pump power density for $RL_B$. (**d**,**e**) The emission spectra of $RL_G$ at different pump power densities when the electric field is removed (**d**) and applied (**e**). (**f**) The integral intensity varies with pump power density for $RL_G$. (**g**,**h**) The emission spectra of $RL_R$ at different pump power densities when the electric field is removed (**g**) and applied (**h**). (**i**) The integral intensity varies with pump power density for $RL_R$. The corresponding spectral integral range is from 440 nm to 470 nm for $RL_B$, from 510 nm to 570 nm for $RL_G$ and from 625 nm to 700 nm for $RL_R$.

Similarly, the effect of scattering feedback of individual scattering part on the emission spectra of green random laser was also studied. A green random laser named as $RL_G$ was obtained by coupling the C153 solution and PDLC film. When the electric field of PDLC film is removed (OFF), coherent feedback random lasing with a central wavelength of about 542.7 nm and a threshold of about 1.15 MW cm$^{-2}$ can be observed (Figure 2d,f). Meanwhile, when the electric field is applied to the PDLC film (ON), coherent feedback random lasing with a threshold of about 0.87 MW cm$^{-2}$ can be obtained (Figure 2e–f). The regulation effect of scattering feedback on emission spectra of $RL_G$ is similar to that of $RL_B$. In addition, the effect of scattering feedback of individual scattering part on the emission spectra of red random laser is demonstrated. A red random laser named as $RL_R$ was achieved by coupling the DCJTB solution and PDLC film. When the electric field of

PDLC film is removed (OFF), coherent feedback random lasing with a threshold of about 1.07 MW cm$^{-2}$ can be observed around 662.9 nm in Figure 2g,i. When the PDLC film is applied electric field and is in ON state, coherent feedback random lasing with a threshold of about 0.87 MW cm$^{-2}$ is obtained (Figure 2h,i). The experimental results show that the random lasing can always be achieved based on the separate coupling configuration with pure gain region and PDLC scattering part. Moreover, the threshold of the blue, green and red random lasing can be effectively modified through regulating the scattering feedback of individual scattering part.

The effect of scattering feedback of individual scattering part on the intensity of random lasing is demonstrated by continuously pumping 100 times at a fixed pump power density in Figure 3. The integral intensities of the emission spectra are calculated when the PDLC film is with external electric field (ON, black sphere) and without electric field (OFF, blue sphere) at a pump power density of 0.62 MW cm$^{-2}$ in Figure 3a. The emission intensity of $RL_B$ in ON state is much higher than that in OFF state, and the average integral intensity of 100 spectra in ON state is about 109 times of that in OFF state. Similarly, $RL_G$ and $RL_R$ are continuously pumped for 100 times under the pump power densities of 1.39 MW cm$^{-2}$ and 1.46 MW cm$^{-2}$, respectively. Their integral intensities of the emission spectra are demonstrated through when the electric field is applied (ON, black sphere) and removed (OFF, blue sphere) in Figure 3b,c. The scattering feedback of individual scattering part has a similar regulation effect on the emission intensity of $RL_G$ and $RL_R$ as $RL_B$. The average integral intensities of $RL_G$ and $RL_R$ in ON state are about 79 and 48 times higher than those in OFF state, respectively. Therefore, the intensity of the random lasing can be dynamically manipulated by applying or removing an external electric field to the PDLC film.

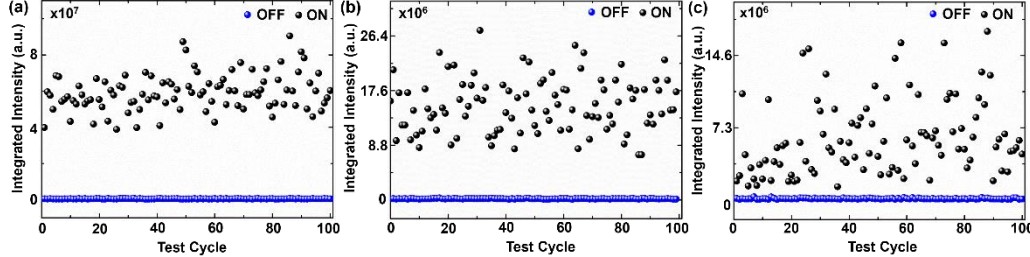

**Figure 3.** Regulation effect of scattering feedback on the intensity of random lasing. (**a**) The integral intensity evolution of the emission spectra when the $RL_B$ is in ON state (black) or OFF state (blue) under a pump power density of 0.62 MW cm$^{-2}$. (**b**) The integral intensity evolution of the emission spectra when the $RL_G$ is in ON state (black) or OFF state (blue) at 1.39 MW cm$^{-2}$. (**c**) The integral intensity evolution of the emission spectra when the $RL_R$ is in ON state (black) or OFF state (blue) at 1.46 MW cm$^{-2}$. The corresponding integral range is (440, 470) nm for $RL_B$, (510, 570) nm for $RL_G$ and (625, 700) nm for $RL_R$.

The effect of scattering feedback of individual scattering part on the spatial coherence of random lasing is also demonstrated by calculating the visibility of interference fringe. The Young's double-slit interference optical path is used to study the spatial coherence of the obtained random lasing. Two slits with a width of 50 μm are separated by 160 μm, and a CCD is positioned behind the slit plane to measure the far-field interference pattern. The visibility ($V$) of the interference fringe in the central region is calculated by $V = (I_{max} - I_{min})/(I_{max} + I_{min})$, where $I_{max}$ and $I_{min}$ are the local maximum and minimum values of the interference intensity near the observation point, respectively [21]. For example, the optical photo and the corresponding intensity distribution of the interference fringe generated by the blue random lasing at the pump power density of 0.35 MW cm$^{-2}$ are shown in Figure 4a,b. The $V$ is about 0.35 for the random lasing related to ON state, and about 0.16 for the random lasing under OFF state. The results show that the spatial coherence of blue random lasing in OFF state is much lower than that in ON state. Similarly, the optical photos and intensity distribution of the interference fringes obtained by the

green random lasing passing through the double-slit interference optical path at the pump power density of 1.32 MW cm$^{-2}$ are shown in Figure 4c,d. The *V* are about 0.34 (ON) and 0.16 (OFF), respectively. In addition, the spatial coherence of the red random lasing is also characterized at 1.32 MW cm$^{-2}$. The *V* of the interference fringes of red random lasing are 0.31 (ON, Figure 4e) and 0.14 (OFF, Figure 4f), respectively. The experimental results show that the visibility of interference fringes of the obtained random lasing in ON state is greater than that in OFF state. Moreover, the spatial coherence of random lasing emitting from another side are also measured, and the results prove that there exists the weak coupling between the gain region and the scattering region of the PDLC film supplying tunable backscattering. Therefore, the spatial coherence of the obtained random lasing can be dynamically manipulated by electrically regulating the scattering coefficient of the PDLC scattering part.

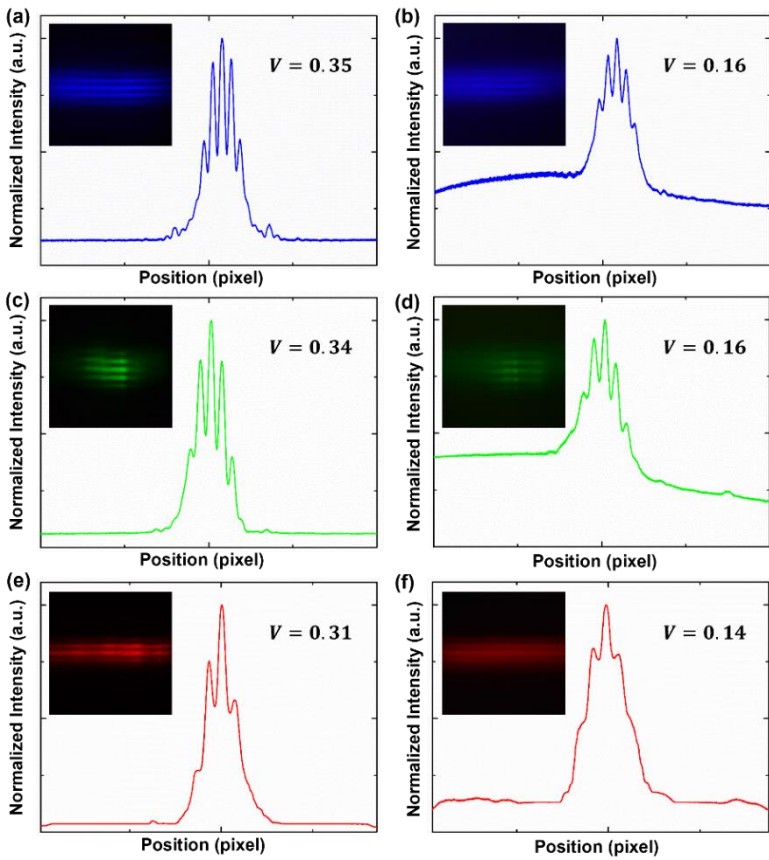

**Figure 4.** Regulation effect of scattering feedback on the spatial coherence of random lasing. (**a**,**b**) The intensity distribution and visibility of the interference fringes obtained by the double slit interference of RL$_B$ under ON state (**a**) or OFF state (**b**). (**c**,**d**) The intensity distribution and visibility of the interference fringes for the RL$_G$ under ON state (**c**) or OFF state (**d**). (**e**,**f**) The intensity distribution and visibility of the interference fringes for the RL$_R$ under ON state (**e**) or OFF state (**f**). Illustrations are the optical photos of the interference fringes, respectively.

## 4. Conclusions

In summary, a random lasing with tunable spatial coherence is designed based on the separate coupling between gain medium and scattering part. The proposed random laser has the advantages of flexible controllability, easy fabrication and recyclability. First, when the PDLC film is separately coupled with different color gain dyes, random lasing with different wavelengths is realized under the excitation of pulsed laser. Then, the scattering intensity of PDLC film is manipulated by applying or removing electric field, and the threshold and intensity of random lasing are regulated. Finally, the spatial coherence of the random lasing can also be dynamically regulated. In short, the intensity and spatial

coherence of random lasing can be manipulated by controlling the electric field. The proposed random lasing is expected to be used in interference measurement, security, optical imaging and display.

**Author Contributions:** Conceptualization, Z.W. and W.G.; Investigation, Y.B. and J.Z.; Methodology, Y.B. and D.L.; Project administration, Z.W.; Supervision, Z.W.; Validation, Z.W.; Visualization, Y.B. and Z.W.; Writing, Y.B., H.Y. and Z.W. All authors have read and agreed to the published version of the manuscript.

**Funding:** This research was funded by National Natural Science Foundation of China (grant Nos. 92150109, 11574033 and 61975018).

**Acknowledgments:** The authors acknowledge support from the National Natural Science Foundation of China and Beijing Higher Education Young Elite Teacher Project.

**Conflicts of Interest:** The authors declare no conflict of interest.

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
