# Peer review of "External Electric Field Tailored Spatial Coherence of Random Lasing"

_crystals, doi:10.3390/cryst12081160_

Round 1

Reviewer 1 Report

Comments in attached file.

Author Response

Dear the reviewer,

  Thanks for your constructive comments and advice. According to your comments, we carefully revised our manuscript. As a result, the quality of the manuscript is improved to a higher level. 

  The attached is a point-to-point response to the comments. Thanks again.

   Sincerely

  Zhaona Wang

Reviewer 2 Report

This paper presents the behavior of a random laser where the gain and scattering media are different and located in separate parts of the system. By using PDLC as scattering medium the authors aim at achieving a control of the spatial coherence of the device. The idea is interesting however there are several issues to be addressed before considering this paper for publication.

1 – The authors should convince the reader that the scattering medium determines the laser emission, that is it makes the devices a random laser rather than a laser plus a variable scattering medium that, obviously, affects the spatial coherence of the laser beam. In fact, it is very easy to get dyes lasing without a real cavity due to the very high gain; actually, the small reflection of a ITO surface may lead to lasing even in presence of a PDLC sample in the ON state (see for instance Lucchetta et al. Appl.Phys.Lett. 84, 837 (2004)). In order to clear this point, it would be necessary to show the emission spectrum of the same device without the PDLC layer but with the ITO surface. Anyway, it is quite difficult to believe that with PDLC in the ON state scattering is still important to determine lasing.

2 – Therefore it is important to show the transmission characteristic of the PDLC layer, that is transmission versus voltage; in fact, it is missing the information about the voltage to be applied to get the ON stats

3 – The basic novelty of the paper should be the data of Figure 4. However, they are very qualitative observations. For instance, the difference in V between Fig.4(e) and Fig.4(f) is rather small. I think more convincing data should be presented about the major claim of the work.

4 – Anyway, I think that FigS1 should be transferred into the main text of the paper.

Author Response

(The authors gave the same response as above.)

Round 2

Reviewer 1 Report

The comments were answered in fully comprehensive.